# Asynchronous Local-SGD Training for Language Modeling

**Bo Liu** [1]   **Rachita Chhaparia** [2]   **Arthur Douillard** [2]   **Satyen Kale** [2]   **Andrei Alex Rusu** [2]   **Jiajun Shen** [2]
**Arthur Szlam** [2]   **MarcAurelio Ranzato** [2]

## Abstract

Local stochastic gradient descent (Local-SGD), also referred to as federated averaging, is an approach to distributed optimization where each device performs more than one SGD update per communication. This work presents an empirical study of *asynchronous* Local-SGD for training language models; that is, each worker updates the global parameters as soon as it has finished its SGD steps. We conduct a comprehensive investigation by examining how worker hardware heterogeneity, model size, number of workers, and optimizer could impact the learning performance. We find that with naive implementations, asynchronous Local-SGD takes more iterations to converge than its synchronous counterpart despite updating the (global) model parameters more frequently. We identify momentum acceleration on the global parameters when worker gradients are stale as a key challenge. We propose a novel method that utilizes a delayed Nesterov momentum update and adjusts the workers' local training steps based on their computation speed. This approach, evaluated with models up to 150M parameters on the C4 dataset, matches the performance of synchronous Local-SGD in terms of perplexity per update step, and significantly surpasses it in terms of wall clock time. Code is available at https://github.com/google-deepmind/asyncdiloco.

## 1. Introduction

Large language models (LLMs) have revolutionized many applications, transforming the way machines interact with human language. The cornerstone of this revolution is training these models at massive scale. To manage such large-scale training in reasonable amounts of time, it has been

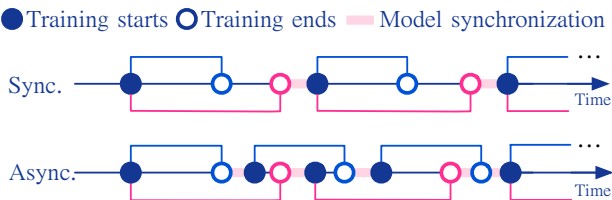

*Figure 1.* Illustration of async. v.s. sync. training with 2 workers (in blue and red). Sync. training suffers from the straggler effect, while async. training reduces the idling time of the fast worker.

necessary to distribute computations across multiple devices. However, the standard approaches to this distributed training uses co-located devices with fast interconnects.

One might hope to be able to effectively harness a broader range of computational resources, perhaps geographically distant from each other, to build even more powerful large models. However, utilizing numerous distant devices faces a significant hurdle: communication latency. When devices focus solely on computing gradients before sending them back to a central server, communication time may exceed computation time, creating an efficiency bottleneck.

Local Stochastic Gradient Descent (Local-SGD) is a collection of optimization methods that can reduce communication bottlenecks.[1] These methods involve each device performing multiple local gradient steps before syncing their parameter updates with a parameter server. While Local-SGD enhances training efficiency by reducing communication frequency, it can suffer from the *straggler effect* caused by heterogeneous devices. For instance, faster devices are idle waiting for slower ones to catch up, undermining the overall efficiency of the system. Moreover, all devices are forced to communicate at the same time requiring high bandwidth connection with the parameter server. Asynchronous Local-SGD presents a more viable solution (illustrated in Figure 1), as it allows the server to update the model as soon as the updates of a worker are available, thereby enhancing computational utilization and minimizing communication bandwidth requirements.

[1]The University of Texas at Austin [2]Google DeepMind. Correspondence to: Bo Liu <bliu@cs.utexas.edu>.

Accepted to the Workshop on Advancing Neural Network Training at International Conference on Machine Learning (WANT@ICML 2024).

[1]The term Local-SGD, sometimes also known as Federated Average (FedAvg), is used here to emphasize its roots in distributed optimization, where users have control over data allocation to different workers.

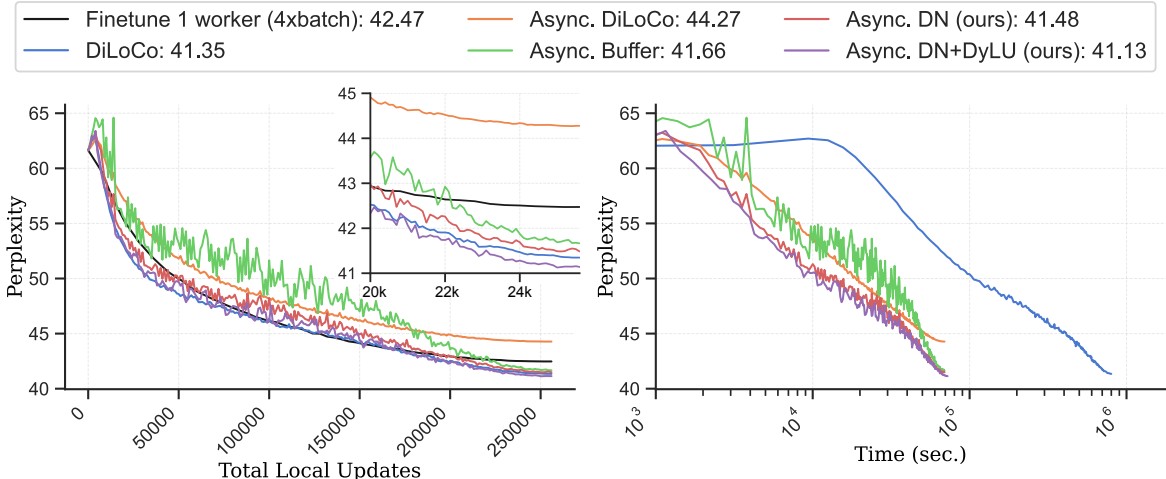

*Figure 2.* Comparative evaluation of language models using sync. and async. Local-SGD methods with 4 heterogeneous workers on a 20M parameter model. The state-of-the-art sync. Local-SGD method, DiLoCo (Douillard et al., 2023), employs `AdamW` and `Nesterov` momentum as the worker-side and server-side optimizers, respectively. This optimizer combination remains the strongest for async. Local-SGD training (See Figure 5), yet underperforms DiLoCo significantly. By integrating Delayed Nesterov (DN) (Algorithm 2) for outer optimization and Dynamic Local Updates (DyLU) (Section 5), we significantly bridge the performance gap in terms of perplexity versus updates between sync. and async. training in language modeling. Moreover, the proposed method significantly surpasses DiLoCo in terms of perplexity versus wall clock time.

In this study, we explore the viability of asynchronously training LLMs using Local-SGD. We expand upon previous works that have attempted to alternate steps on subsets of workers or randomly drop certain subset of workers during synchronous Local-SGD (Ryabinin et al., 2021; Douillard et al., 2023). The main content is structured into three parts:

**1. Framework (Section 3).** The first part introduces our high-level design for the asynchronous training framework. We discuss how each worker determines which data shard to train on, for how many steps, with what learning rates, and how the server updates models asynchronously.

**2. Optimization Challenge (Section 4).** In the second part, we conduct an empirical study of various existing optimization strategies suitable for asynchronous Local-SGD. This includes both worker-side optimization (inner optimization) and server-side optimization (outer optimization). We uncover a key challenge in utilizing momentum effectively. Notably, while adaptive momentum methods accelerate convergence of both inner and outer optimizations, their efficacy in asynchronous Local-SGD is comparatively reduced when both optimizations employ momentum techniques, when contrasted with the synchronous implementation.

**3. Proposed Solutions (Section 5).** We introduce two simple and effective techniques: the Delayed Nesterov momentum update (DN) and Dynamic Local Updates (DyLU). These techniques, when combined and evaluated on training language model, allow asynchronous Local-SGD to approach synchronous Local-SGD in terms of perplexity versus the total number of local updates, and further improve

asynchronous Local-SGD vs. synchronous Local-SGD in terms of perplexity versus wall-clock (Figure 2).

## 2. Background

In this study, we focus on the distributed optimization of shared model parameters $\theta$ across $k$ data shards, denoted as $\mathcal{D} = \{\mathcal{D}_1, \ldots, \mathcal{D}_k\}$, with $k$ workers.[2] The primary goal is described by the following equation:

$$\min_\theta \sum_{i=1}^k \frac{|\mathcal{D}_i|}{\sum_j |\mathcal{D}_j|} \mathbb{E}_{x \sim \mathcal{D}_i} \big[\ell(x; \theta)\big], \qquad (1)$$

where $\ell(\cdot; \theta)$ represents the loss function (for instance, cross-entropy loss for next token prediction in language modeling), and $|\cdot|$ indicates the set size.

We extend the definition of Local-SGD in this work to include not just the original Local-SGD method, but also its variants that incorporate advanced optimization techniques. We particularly focus on DiLoCo (Douillard et al., 2023), which sets the standard for synchronous Local-SGD in language modeling. DiLoCo's methodology is detailed in Algorithm 3. Each worker $i$ performs $H$ local updates using an *inner optimizer* on their data shard $\mathcal{D}_i$ before sending the parameter change (pseudo-gradient) $\delta_i^{(t)} = \theta^{(t-1)} - \theta_i^{(t)}$ back to the server. The server then computes the aggregated outer gradient $\Delta^{(t)} = \frac{1}{k} \sum_{i=1}^k \delta_i^{(t)}$, and applies an *outer op-*

---

[2]We assume the number of workers ($k$) equals the number of data shards, though our methods are also applicable when there are fewer workers than data shards.

*timizer* with $\Delta^{(t)}$ to update $\theta$. A key insight from DiLoCo is the optimal use of `AdamW` and `Nesterov Momentum` as the best inner and outer optimizers, respectively.

## 3. Async. Local-SGD Framework

This section outlines the asynchronous Local-SGD pipeline design, where we assume a central server controls all workers and asynchronously aggregates their updates.

**Data Shard Sampling** Unlike in the federated learning setting where each device is attached to its own data, in distributed optimization, the user has the right to choose which data shard is assigned to which worker, even dynamically. To balance the learning progress on different data shards (as workers are heterogeneous), whenever a worker is ready to start a new local optimization round, we sample a data shard inversely proportional to its "learning progress". Specifically, define $n_i$ as the number of learned data points in $\mathcal{D}_i$, then we sample a shard $i_{\text{sampled}}$ according to:

$$i_{\text{sampled}} \sim p,$$
$$\text{where } p_i \propto \max\left(\frac{|\mathcal{D}_i|}{\sum_j |\mathcal{D}_j|} - \frac{n_i}{\sum_j n_j}, 0\right). \quad (2)$$

In other words, we sample a data shard only when it is "under-sampled" (i.e., $\frac{n_i}{\sum_j n_j} \leq \frac{|\mathcal{D}_i|}{\sum_j |\mathcal{D}_j|}$). The degree to which a shard is under-sampled determines its sampling rate. By doing so, we ensure that the data shard with slower progress is more likely to be sampled for training, therefore balancing the learning progress across shards.

**Learning Rate Scheduling** In contrast to synchronous training methods like DiLoCo, asynchronous training can lead to uneven progress across different data shards, especially when workers are allowed varying numbers of training steps. This raises the question of how to effectively schedule learning rates. In our approach we assign each data shard its own learning rate schedule. Specifically, we implement a linear warmup combined with a cosine learning rate decay, where $T$ represents the target total training iterations for each data shard:

$$\eta_t = \begin{cases} t\eta_{\text{max}}/t_{\text{warmup}} & t < t_{\text{warmup}} \\ \eta_{\text{min}} + 0.5(\eta_{\text{max}} - \eta_{\text{min}}) & \\ \quad \left(1 + \cos\left(\frac{t-t_{\text{warmup}}}{T-t_{\text{warmup}}}\pi\right)\right) & t \geq t_{\text{warmup}}. \end{cases} \quad (3)$$

In practice, asynchronous training may conclude with different final iteration counts ($t_{\text{end}}$) for each data shard. Since we cannot predetermine $t_{\text{end}}$ due to the unpredictability of asynchrony, we set the minimum learning rate ($\eta_{\text{min}}$) to a small positive value. This ensures continued progress even if $t$ exceeds $T$. Additionally, we adjust $T - t_{\text{warmup}}$ to be non-negative and ensure that the ratio $\frac{t-t_{\text{warmup}}}{T-t_{\text{warmup}}}$ remains within the range of $[0, 1]$. This helps maintain effective learning rate adjustments throughout the training process.

**Grace Period for Model Synchronization** In asynchronous training, the completion time of each worker's tasks can vary. For example, if worker B completes training shortly after worker A, it might be beneficial for A to wait briefly until the server processes updates from both workers before receiving the updated model for its next training task. However, this waiting period should be minimal and occur only when necessary. Specifically, if no other worker completes its task within the grace period while worker A is synchronizing with the server's model, A should promptly commence its new training task using the server's current model. For a visual representation of this process, please refer to Figure 3.

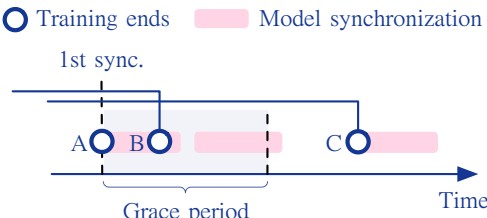

*Figure 3.* We consecutively synchronize the update from B after we synchronize A because B finishes training after A but before the end of the grace period. So A and B use the same server model to start new training jobs, while C will start its own grace period.

**Asynchronous Task Scheduling** In Algorithm 1, we present the asynchronous task scheduling pipeline. Throughout the algorithm, we use $\tau$ to denote the actual wall clock time and $t$ to denote model updates. In **line 1-4**, we initialize the model, total local updates $t_{\text{local}}$, and the list of workers $\mathcal{W}$ and the completed workers $\mathcal{W}_{\text{completed}}$. In **line 5**, we start the first training job for all workers with the initial model parameter $\theta^{(0)}$. Note that the train() function implements the data sampling technique and performs the learning rate scheduling mentioned before. In **line 6**, we reset the starting time of the grace period $\tau_{\text{sync}}$ to $\infty$. This is because we want to synchronize with a worker only when its completion time is within $\tau_{\text{sync}} + \tau_{\text{grace}}$. The main asynchronous Local-SGD training loop is provided in **line 6-19**. Within the loop, we first attempt to get a completed worker $w$ (**line 7**). We retrieve the earliest completed worker that we have not yet processed yet, as long as its completion time is still within the grace period (e.g., $w.\text{completed\_time} \leq \tau_{\text{sync}} + \tau_{\text{grace}}$). If no such workers exist, get_worker() will return null. In **line 10-15** where such a worker $w$ is found, we synchronize its update with the server model $\theta$. In **line 17-20** when no such workers are found, we assign new training jobs for all completed workers and empty the list of completed workers. For the detailed pseudocode of the train() and get_worker() functions, please refer to Appendix E. In practice, for the sake of reproducibility of research, we implement a *deterministic* version of Algorithm 1 with faked training time based on real-world device statistics. We validate the cor-

**Algorithm 1** Async. Local-SGD Task Scheduling.

```
 1: Require: Initial pretrained model θ⁽⁰⁾
 2: Require: k workers
 3: Require: Grace period τ_grace
 4: Require: Total local updates t_max
 5: t_local = 0
 6: θ ← θ⁽⁰⁾
 7: W = [init_worker() for i in [k]]
 8: W_completed = []
 9: train(W, θ)    // the initial round of training
10: τ_sync = ∞    // start of the grace period
11: while t_local < t_max do
12:     // get a completed worker
13:     w = get_worker(W, τ_grace, τ_sync)
14:     if w exists then
15:         // synchronize the update with server
16:         τ_sync = min(τ_sync, w.completed_time)
17:         θ ← sync(θ, w.update)
18:         W_completed.add(w)
19:         t_local += w.local_updates
20:     else
21:         // assign jobs for completed workers
22:         τ_sync = ∞
23:         train(W_completed, θ)
24:         W_completed = []
25:     end if
26: end while
```

rectness of the training pipeline by simulating synchronous updates using the asynchronous framework.

## 4. Optimization Challenge

**Effect of `InnerOpt + OuterOpt`**   To study how optimization affects the language modeling performance in asynchronous Local-SGD, we first experiment with different combinations of the inner and outer optimizers (we use A+B to denote A and B as the inner and outer optimizer, respectively): `SGD+Nesterov`, `SGD+Adam`, `AdamW+SGD`, `AdamW+SGD Momentum`, `AdamW+Adam`, `AdamW+Nesterov`. The hyperparameters for each combination are tuned separately, for AdamW as `InnerOpt` we kept the default values. We assume there are $k = 4$ workers, whose device speed is shown in Figure 4. Then we apply asynchronous Local-SGD finetuning on a 20M-parameter language model for 64,000 steps per worker (256,000 local steps in total), where the initial model checkpoint was pretrained for 24,000 steps with `Adam` without distributed training. We choose finetuning with Local-SGD as it has been observed that Local-SGD style methods work well in finetuning but is less efficient from scratch (Lin et al., 2018), though others have also observed that Local-SGD works well even for training from scratch (Douillard et al., 2023).

The learning rate scheduling and task scheduling follow the procedures described in Section 3. We use inner steps = 50 across all workers in all experiments by default. The result is shown in Figure 5.

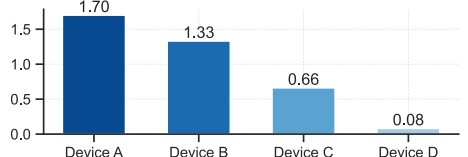

*Figure 4.* Steps per second for each device.

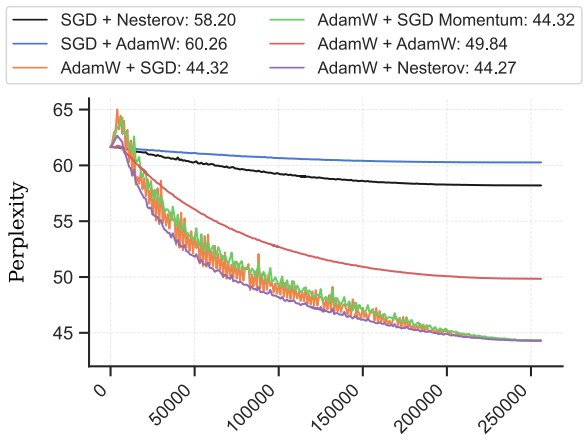

*Figure 5.* Performance of using different combinations of inner and outer optimizers for asynchronous Local-SGD training on a 20M language model with 4 workers.

**Observation**   The analysis reveals that combining `AdamW` as the inner optimizer with `Nesterov` momentum as the outer optimizer yields the best results, aligning with findings from synchronous training, like the DiLoCo method. Notably, using `AdamW` as the outer optimizer is less effective. This may be because `AdamW`, derived from `Adam`, introduces a normalization effect, which can be counterproductive in Local-SGD where pseudo-gradients tend to be larger than true gradients, potentially slowing convergence. When `AdamW` is used in the inner optimization, `SGD`, `SGD Momentum`, and `Nesterov` show comparable performance. However, `Nesterov` not only stabilizes the learning curve but also slightly improves final performance. This can be attributed to its update mechanism (here we abuse the notation and let $t$ denote $t_{\text{server}}$):

$$
\begin{aligned}
m_{t+1} &= \beta m_t + g_t \\
\theta_{t+1} &= \theta_t - \epsilon\big(\beta^2 m_t + (1+\beta)g_t\big),
\end{aligned}
\tag{4}
$$

where $\epsilon$ is the learning rate, $m_t$ is the momentum, $g_t$ the gradient at time $t$, and $\beta \in (0, 1)$ the decay factor. The key difference between `Nesterov` and `SGD Momentum` is in how `Nesterov` adjusts the weightings, reducing the

momentum component ($\beta^2$ instead of $\beta$) and increasing the gradient component ($1 + \beta$ instead of 1). This suggests that momentum plays a crucial yet intricate role in Local-SGD.

**Momentum in the `OuterOpt`** To delve deeper into the momentum term's impact on the outer optimizer, we conducted comparative analyses between `AdamW+SGD` and `AdamW+Nesterov` under both synchronous and asynchronous training settings. These experiments were carried out under identical conditions as described earlier. The results are reported in Figure 6.

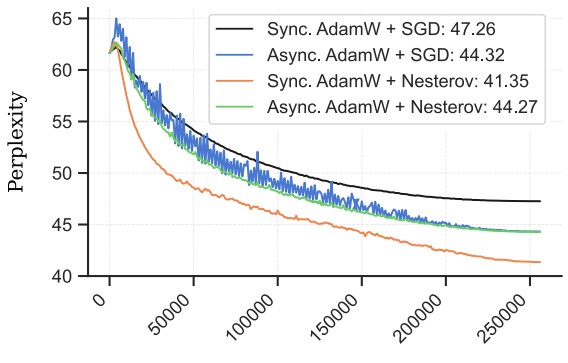

*Figure 6.* Comparison of `AdamW+SGD` and `AdamW+Nesterov` in both synchronous and asynchronous Local-SGD training.

**Observation** The figure clearly shows that in asynchronous Local-SGD, `AdamW+SGD`, which lacks a momentum term, leads to better final perplexity and learning efficiency than its synchronous counterpart. However, incorporating `Nesterov` momentum into the `OuterOpt` significantly boosts the performance of synchronous Local-SGD, outperforming the asynchronous version. It's noteworthy that asynchronous `AdamW+Nesterov` remains the best performer across all tested combinations of inner and outer optimizers (as seen in Figure 5). This observation indicates that while momentum is beneficial in asynchronous Local-SGD for language modeling, its effect is more pronounced in synchronous settings.

**Is Staleness the Cause?** We further apply the asynchronous DiLoCo algorithm with homogeneous devices. By doing so, we maximally diminish the staled gradient problem in Local-SGD, which refers to the fact that we are using an outdated outer gradient to update the server model. In particular, this means if we have $k$ workers, all of them will return the computed outer gradient back to the server at the same time. Therefore, the only staleness comes from the fact that we are sequentially applying the individual updates instead of aggregating them together and apply it once. Results are summarized in Figure 7.

**Observation** Figure 7 reveals a notable finding: even with homogeneity among workers, asynchronous DiLoCo sig-

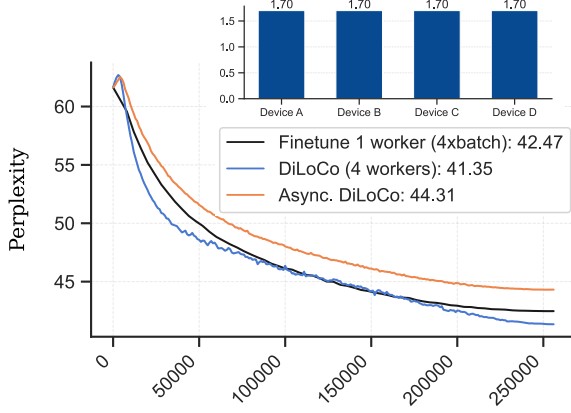

*Figure 7.* Async. DiLoCo with heterogeneous devices.

nificantly lags behind its synchronous counterpart. This suggests that the *inherent staleness* from sequentially applying simultaneous updates leads to considerable performance drops. To elucidate this effect, let's consider a scenario with $k = 4$ workers providing identical outer gradients (denoted as $g$). The standard Nesterov momentum update is outlined in Equation (4). In a sequential application of pseudo-gradients:

$$
\begin{aligned}
m_{t+1} &= \beta^4 m_t + (1 + \beta + \beta^2 + \beta^3)g \\
\theta_{t+1} &= \theta_t - \epsilon\big((4 + 4\beta + 3\beta^2 + 2\beta^3 + \beta^4)g \\
&\quad + \beta^2(1 + \beta + \beta^2 + \beta^3)m_t\big).
\end{aligned} \quad (5)
$$

From this, we observe that sequential application results in a more rapidly decaying momentum term but amplifies the actual change in parameter $\theta$. Consequently, a higher $\beta$ maintains more recent momentum but may lead to greater changes in parameters, and vice versa. Note this imbalance cannot be easily removed by reducing the learning rate.

**Baselines** We consider several synchronous baselines from the literature, and their naive application to an asynchronous setting: **1)** Finetune 1 worker (4xbatch): This involves finetuning a single worker with a larger batch size, equating to synchronous SGD. **2)** DiLoCo (Douillard et al., 2023): This synchronous Local-SGD method combines `AdamW` with `Nesterov`. **3)** Async. DiLoCo: The asynchronous version of DiLoCo.

**Existing Fixes** We investigated potential fixes from the asynchronous Local-SGD literature to address observed challenges. The following methods were considered: **1)** Async. DiLoCo + Poly (Xie et al., 2019): Extends Async. DiLoCo by downweighting the pseudo-gradient with $g \leftarrow (1 + \text{staleness})^{-0.5}g$. **2)** Async. DiLoCo + PolyThres: Adds a threshold to discard updates with staleness beyond 10. **3)** Async. DiLoCo + Delay Comp. (Zheng et al., 2017): Introduces delay compensation (Delay Comp.) to approximate true pseudo-gradients. Denote the gradient function

at $\theta$ as $g(\theta)$, then the main idea of delay compensation is to approximate the true gradient $g(\theta_t)$ by a stale gradient $g(\theta_{t'})$ ($t' < t$) with the first-order Taylor approximation, e.g., $g(\theta_t) \approx g(\theta_{t'}) + \nabla g(\theta_{t'})(\theta_t - \theta_{t'})$. In practice, the Hessian $\nabla g(\theta_{t'})$ is approximated by diagonalized gradient outer product, e.g., $\nabla g(\theta_{t'}) \approx \lambda g(\theta_{t'}) \odot g(\theta_{t'})$, where $\lambda \in \mathbb{R}^+$ is a hyperparameter. In our setting, we apply the delay compensation technique to pseudogradients instead of gradients. **4) Async. Buffer:** Accumulates and averages all gradients in a First-In, First-Out fashion before applying Nesterov updates; a variation of the original FedBuff algorithm (Nguyen et al., 2022), using `AdamW+Nesterov`. The results are provided in Figure 8.

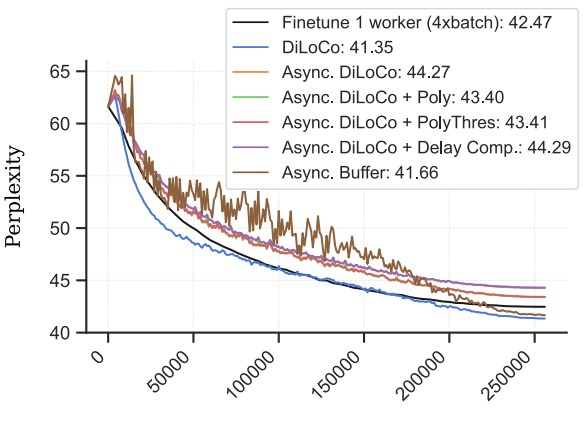

*Figure 8.* Comparison of different asynchronous Local-SGD approaches. Poly, PolyThres, and Delay Comp. barely improve the async. Local-SGD performance. Async. Buffer significantly closes the gap between sync. and async. training, while introducing instability in early stage of training.

**Observation** Polynomial discounting of the pseudo-gradient shows marginal benefits. Thresholding and delay compensation techniques don't offer much improvements. Again, the fact that delay compensation is not working well points out the difference between asynchronous SGD and asynchronous Local-SGD. The Async. Buffer method excels at convergence but exhibits instability early in training. Crucially, *none* of the methods match the performance of the synchronous DiLoCo baseline.

**A Minimal Toy Example** To facilitate future research and expedite the prototyping of novel ideas, we provide a minimal toy example (a one-file Python code), that reproduces the optimization challenge observed in asynchronous Local-SGD (refer to Appendix C)

## 5. Proposed Solutions

In addressing the optimization challenges outlined in Section 4, we developed two strategies.

**Delayed Nesterov Update** Notably, the Async. Buffer method demonstrated promising performance (as shown

---

**Algorithm 2** Delayed Nesterov Update.

1: **Require:** Initial model parameter $\theta_0$
2: **Require:** Momentum decay $\beta \in (0,1)$
3: **Require:** Momentum activation $c \in [0, 1/N]$
4:                                                            // default to $c = 0$
5: **Require:** Buffer size $N$
6: $t = 0$
7: $m_0 = 0$     // momentum
8: $\Delta = 0$     // aggregated gradient
9: **while** not finished **do**
10:     Receive the pseudo-gradient $g_t$
11:     // sync. step in Alg. 1.
12:     $\Delta \leftarrow \Delta + g_t$
13:     **if** $(t + 1) \% N == 0$ **then**
14:         $m_{t+1} \leftarrow \beta m_t + \Delta/N$
15:         $\theta_{t+1} \leftarrow \theta_t - \epsilon\big((1 - cN + c)\beta m_{t+1} + g_t/N\big)$
16:         $\Delta = 0$
17:     **else**
18:         $m_{t+1} \leftarrow m_t$     // delay momentum update
19:         $\theta_{t+1} \leftarrow \theta_t - \epsilon\big(c\beta m_{t+1} + g_t/N\big)$
20:     **end if**
21:     $t \leftarrow t + 1$
22: **end while**

---

in Figure 8). Additionally, our analysis revealed that asynchronous training with `AdamW+SGD`, sans outer momentum, outperforms synchronous methods (Figure 5). Based on these insights, we propose the *Delayed Nesterov* (DN) strategy, which represents the sync() function in Algorithm 1. This approach involves using the `Nesterov` update intermittently—every $N$ server updates. Between `Nesterov` updates, we aggregate pseudo-gradients in a buffer $\Delta$ and update the model parameters using gradient descent (or gradient descent plus a small fraction of the old momentum). To balance gradient and momentum-based descent, we introduce a parameter $c \in [0, 1/N]$. A $c$ value of 0 indicates pure gradient descent between `Nesterov` updates, while $c$ equal to 1 evenly distributes the momentum term over $N$ updates. The specifics of this algorithm are detailed in Algorithm 2. Unlike the Async. Buffer (Nguyen et al., 2022), which updates model parameters only once in $N$ periods, the Delayed Nesterov continuously updates using gradients, incorporating a fraction of the old momentum, and updates the momentum term once every $N$ server updates.

**Dynamic Local Updates** The Delayed Nesterov approach addresses the momentum challenge in the `OuterOpt` by buffering pseudo-gradients and strategically delaying momentum updates. An alternative perspective considers synchronous training as a solution, where pseudo-gradients from all workers are synchronized. However, the diversity in device capabilities often hinders simultaneous pseudo-gradient returns, if each worker executes the same number

of local training steps. A viable workaround involves customizing local training steps (e.g., $w$.steps) based on the processing speed of each device. In particular, denote $v(w)$ as the training speed (in terms of steps per second) for worker $w$, we can compute a worker's desired training steps as:

$$w.\text{step} = \left\lfloor \frac{v(w)}{\max_{w' \in \mathcal{W}} v(w')} H \right\rfloor, \qquad (6)$$

where $H$ denotes the number of local training steps the fastest worker runs and $\lfloor x \rfloor$ denotes the largest integer not greater than $x$.[3] We name this approach the Dynamic Local Updates (DyLU). This adjustment allows slower workers to execute fewer steps, aligning the completion times across different workers. Incorporating a grace period for model synchronization in this setup further reduces the impact of stale gradients, improving overall performance.

# 6. Experiments

This section details experiments conducted to assess the efficacy of our two proposed methods, Delayed Nesterov (DN) and Dynamic Local Updates (DyLU). Additionally, ablation studies explore the effectiveness of these methods as we vary the number of workers and model sizes.

**Evaluating Delayed Nesterov (DN) + Dynamic Local Updates (DyLU)**  Figure 2 compares async. Local-SGD with DN+DyLU against baselines such as single worker finetuning and DiLoCo, using the same setup as in Figure 8.

**Observation**  The results demonstrate that DN combined with DyLU significantly reduces perplexity, surpassing the synchronous DiLoCo's performance over updates. Additionally, DN+DyLU outperforms in terms of time efficiency, avoiding delays from slower workers.

**Assessing Different Levels of Worker Heterogeneity**  We examine how both the proposed DN+DyLU method and vanilla asynchronous DiLoCo fare under varying degrees of worker device heterogeneity, as shown in Figure 9 (perplexity curve) and Table 1 (final perplexity).

*Table 1.* Varying the level of worker heterogeneity (**top-left**, **top-right**, **bottom-left**, and **bottom-right** of Figure 9 correspond to **no**, **slight**, **moderate**, and **very**, respectively).

| Level of heterogeneity | no | slight | moderate | very |
|---|---|---|---|---|
| Pretrained (24K) | 61.64 | 61.64 | 61.64 | 61.64 |
| Finetune ($4\times$ batch size) | 42.47 | 42.47 | 42.47 | 42.47 |
| DiLoCo (Douillard et al., 2023) | 41.35 | 41.35 | 41.35 | 41.35 |
| Async. DiLoCo | 44.27 | 44.38 | 44.29 | 44.27 |
| Async. DN + DyLU (ours) | **41.27** | **41.27** | **41.09** | **41.13** |

**Observation**  DN+DyLU consistently excels across all heterogeneity levels.[4] Interestingly, even with homogeneous devices, vanilla asynchronous DiLoCo struggles, suggesting that the issue partly lies in the sequential application of pseudogradients. This underscores the importance of delayed momentum updates. Additionally, a periodic oscillation in performance is observed in certain device groupings, further highlighting the lack of robustness of the original asynchronous algorithm.

**Ablation with Different Numbers of Workers**  We apply DN+DyLU while varying the number of workers (4, 8, 16) using a 20M model, with results summarized in Figure 10 (perplexity curve) and Table 2 (final perplexity).

*Table 2.* Varying the number of workers.

| Number of workers $k$ | 4 | 8 | 16 |
|---|---|---|---|
| Pretrained (24K) | 61.64 | 61.64 | 61.64 |
| Finetune ($k\times$ batch size) | 42.47 | 41.28 | **40.60** |
| DiLoCo (Douillard et al., 2023) | 41.35 | 41.23 | 41.25 |
| Async. DiLoCo | 44.27 | 44.23 | 44.23 |
| Async. DN + DyLU (ours) | **41.13** | **41.02** | 40.98 |

**Observation**  As the number of workers increases, the benefit of Local-SGD training diminishes. Notably, with 16 workers, single worker finetuning (16x batch size) shows the best performance over updates. Yet, DN+DyLU closely aligns with synchronous DiLoCo in performance, demonstrating its potential as a DiLoCo alternative in heterogeneous settings.

**Ablation with Different Model Sizes**  Lastly, we apply DN+DyLU to models of varying sizes (20M, 60M, 150M), with results summarized in Figure 11 (perplexity curve) and Table 3 (final perplexity).

*Table 3.* Varying the model sizes.

| Model size | 20M | 60M | 150M |
|---|---|---|---|
| Pretrained (24K) | 61.64 | 30.19 | 22.80 |
| Finetune (4x batch size) | 42.47 | 24.80 | 17.47 |
| DiLoCo (Douillard et al., 2023) | 41.35 | 24.55 | **17.23** |
| Async. DiLoCo | 44.27 | 25.64 | 18.08 |
| Async. DN + DyLU (ours) | **41.13** | **24.53** | 17.26 |

**Observation**  Both synchronous and asynchronous Local-SGD methods outperform the approach of finetuning a single worker with an increased batch size. Notably, this advantage becomes more pronounced during the later stages of convergence, aligning with findings from previous research that highlight Local-SGD's superior generalization capabilities (Gu et al., 2023). Additionally, our proposed

---

[3]We implicitly assume the device speeds are given. Otherwise, it is straightforward to estimate the device speed empirically.

[4]We notice that Async. DN+DyLU performs slightly better than DiLoCo when there is no heterogeneity, this is due to the numerical error, as the training curves match almost perfectly.

DN+DyLU method demonstrates consistent efficacy across various model sizes. It's important to note that the performance disparity between synchronous and asynchronous DiLoCo does not diminish even as the model size increases.

**Ablation with Different** $c$ We apply $c \in \{0, 0.1\}$ in Async. DN+DyLU with varying $k$ (4, 8, 16) and model sizes (20M, 60M, 150M), with the 4 "very" heterogeneous workers. This is because when the level of heterogeneity is small, using different $c$ will have smaller difference (e.g., when there is no heterogeneity, any $c$ results in the same algorithm). Results are summarized in Table 4.

*Table 4.* Varying the $c \in \{0, 0.1\}$ in Algorithm 2.

| Number of workers $k$ | 4 | 8 | 16 |
|---|---|---|---|
| Async. DN + DyLU ($c = 0$) | **41.13** | 41.02 | **40.98** |
| Async. DN + DyLU ($c = 0.1$) | 41.16 | **40.93** | 41.04 |
| Model size | 20M | 60M | 150M |
| Async. DN + DyLU ($c = 0$) | **41.13** | **24.53** | **17.26** |
| Async. DN + DyLU ($c = 0.1$) | 41.16 | 24.69 | 17.27 |

**Observation** Empirically, we observe no significant difference between $c = 0$ and $c = 0.1$, indicating that adding slight momentum at intermediate steps does not help too much. As a result, we set $c = 0$ as the default value in Algorithm 2, which corresponds to performing SGD updates between two consecutive Nesterov updates. Note that setting the value of $c$ does not introduce any overhead to the overall algorithm.

# 7. Related Work

This section provides a concise overview of the literature on federated learning and local-SGD style distributed optimization, particularly focusing on the asynchronous settings.

**Local-SGD and Distributed Optimization** Local-SGD is a specific distributed optimization technique designed to reduce communication frequency (Stich, 2018; Zhang et al., 2016; Bijral et al., 2016; McDonald et al., 2010; Coppola, 2015; Zinkevich et al., 2010). The core principle of Local-SGD is to let each worker execute several local training iterations before engaging in global synchronization. This technique was later applied to the federated learning setting, leading to the development of the FedAvg method (McMahan et al., 2017), which aims to reduce communication overhead. Unlike Local-SGD, federated learning also addresses user privacy issues and typically involves heterogeneous devices. To further minimize communication overhead, FedOpt integrates adaptive optimization methods like SGD momentum and Adam (Reddi et al., 2020). However, as worker heterogeneity increases, the convergence rate often deteriorates. Methods like SCAFFOLD (Karimireddy et al., 2020) and MIME (Karimireddy et al., 2021) are introduced

to adapt these methods for heterogeneous environments.

**Asynchronous Training** Asynchronous training was developed to mitigate the "straggler effect" observed in synchronous distributed optimization, where learning efficiency is bottlenecked by the slowest worker (Koh et al., 2006; Recht et al., 2011; Dean et al., 2012; Lian et al., 2015; 2018; Diskin et al., 2021b). A significant challenge in asynchronous optimization is the staled gradient problem, which occurs when an outdated gradient is applied to a recently updated model. Asynchronous SGD with delay compensation (Zheng et al., 2017) addresses this issue by approximating the true gradient using the old gradient. Asynchronous methods have also been explored in federated learning contexts (Xie et al., 2019). Despite the challenge, asynchronous training has demonstrated success for language modeling as well (Diskin et al., 2021b), by using devices world-wide.

**Local-SGD for Language Modeling** The concept of local-SGD (or FedAvg) has previously been applied in the realm of language modeling. Cross-device federated learning, for instance, has been utilized to pretrain and fine-tune language models (Hilmkil et al., 2021; Ro et al., 2022; Ryabinin et al., 2021; Diskin et al., 2021a; Presser, 2020; Borzunov et al., 2022). More recently, DiLoCo has extended the local-SGD methodology to encompass larger language models, specifically proposing the use of AdamW + Nesterov momentum as the `InnerOpt` + `OuterOpt` pairing. In asynchronous settings, the FedBuff (Nguyen et al., 2022) algorithm buffers pseudogradients from clients, updating the server model only after accumulating a sufficient number of pseudogradients. TimelyFL (Zhang et al., 2023) aims to reduce asynchrony by allowing slower devices to train only parts of the model.

# 8. Conclusions and Limitations

This study examines asynchronous Local-SGD in language modeling, identifying that while momentum is key in the outer optimization loop, its effectiveness decreases in asynchronous settings compared to synchronous ones when applied simplistically. To address this, we propose a novel method using sporadic momentum updates with buffered pseudogradients alongside continuous stochastic pseudogradient updates. We find that adjusting local training steps to match each worker's computational speed significantly boosts performance. This study, while thorough, has several limitations. The exact cause of the optimization challenge is uncertain, highlighting a need for further theoretical research. Additionally, the effectiveness of the Local-SGD method appears to decrease as the number of workers increases, an issue that affects its scalability and requires more understanding. Lastly, despite DN+DyLU's enhanced performance, it lacks a theoretical convergence analysis, warranting additional exploration.

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

## A. Synchronous Local-SGD (DiLoCo (Douillard et al., 2023)) Pseudocode

Here, we provide the pseudocode for the synchronous Local-SGD training pipeline in Algorithm 3, using the optimization suggested by DiLoCo (Douillard et al., 2023).

---

**Algorithm 3** DiLoCo Algorithm (synchronous)

---

1: **Require:** Initial pretrained model $\theta^{(0)}$
2: **Require:** $k$ workers
3: **Require:** Data shards $\{\mathcal{D}_1, \ldots, \mathcal{D}_k\}$
4: **Require:** Optimizers `InnerOpt` and `OuterOpt`
5: **for** `outer step` $t = 1 \ldots T$ **do**
6:    **for** `worker` $i = 1 \ldots k$ `in parallel` **do**
7:       $\theta_i^{(t)} \leftarrow \theta^{(t-1)}$
8:       **for** `inner step` $h = 1 \ldots H$ **do**
9:          $x \sim \mathcal{D}_i$
10:          $\mathcal{L} \leftarrow f(x, \theta_i^{(t)})$
11:          $\theta_i^{(t)} \leftarrow \texttt{InnerOpt}(\theta_i^{(t)}, \nabla_{\mathcal{L}})$
12:       **end for**
13:       $\delta_i^{(t)} = \theta^{(t-1)} - \theta_i^{(t)}$
14:    **end for**
15:    $\Delta^{(t)} \leftarrow \frac{1}{k} \sum_{i=1}^{k} \delta_i^{(t)}$    // outer gradients
16:    $\theta^{(t)} \leftarrow \texttt{OuterOpt}(\theta^{(t-1)}, \Delta^{(t)})$
17: **end for**

---

## B. Additional Results

We provide the plots of the performance (perplexity) of various methods for different ablation studies mentioned in Section 6 in the following.

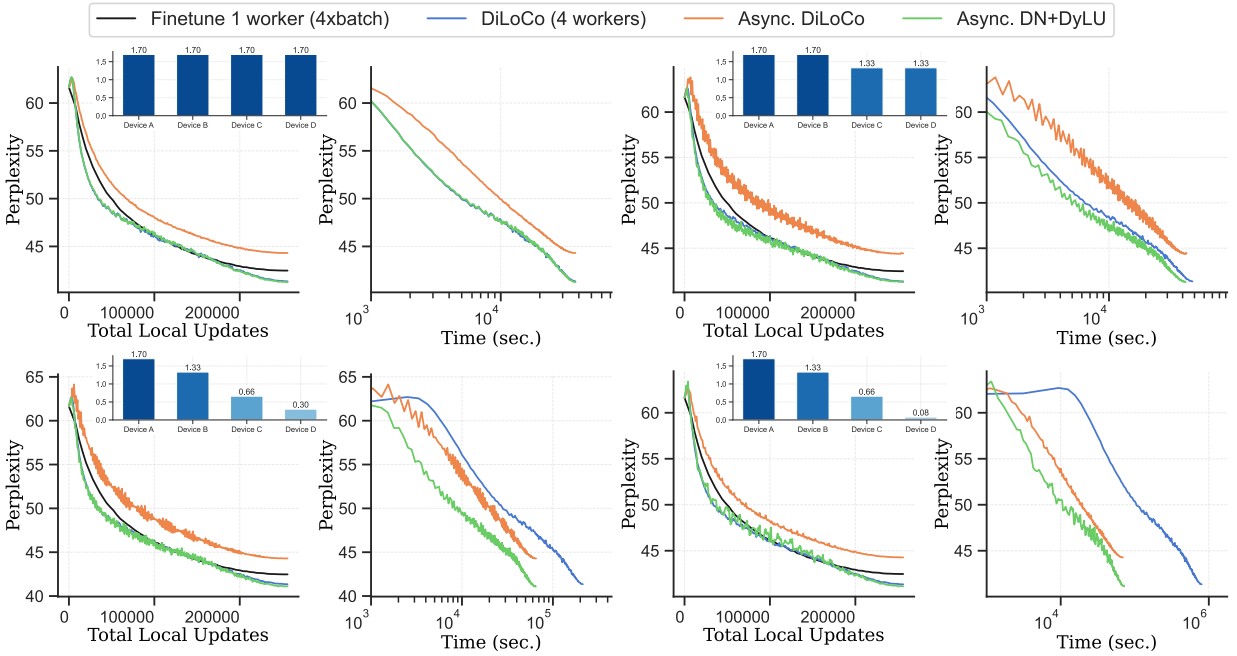

*Figure 9.* Varying the heterogeneity in devices.

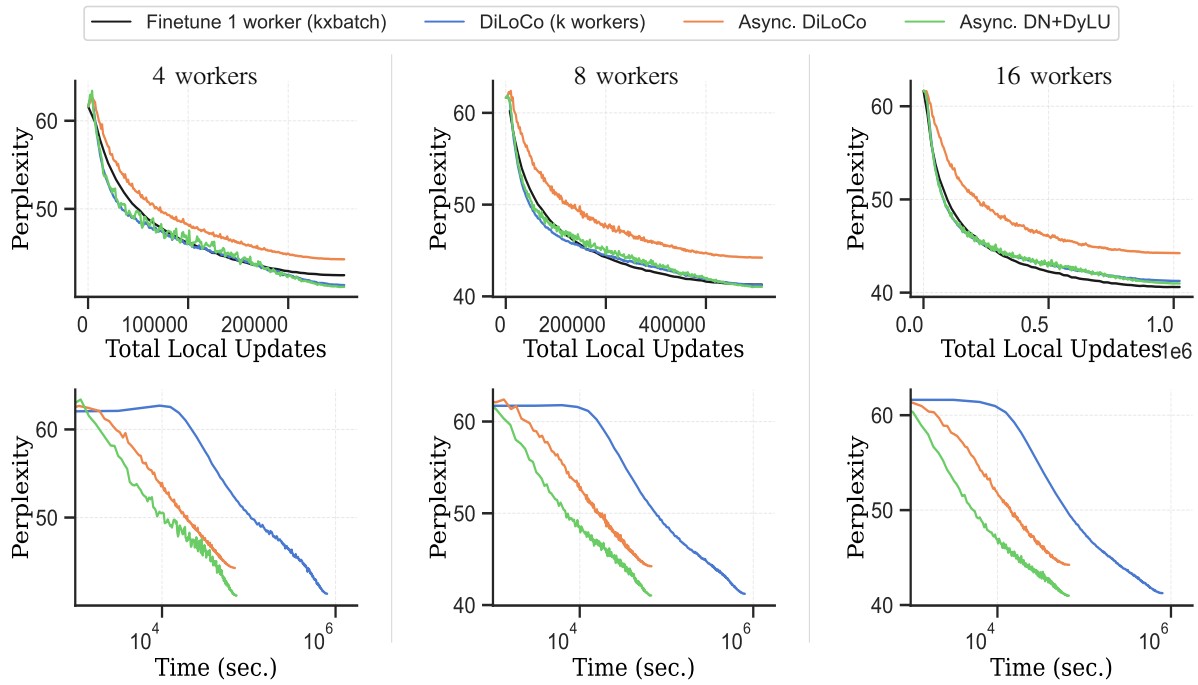

*Figure 10.* Varying the number of workers.

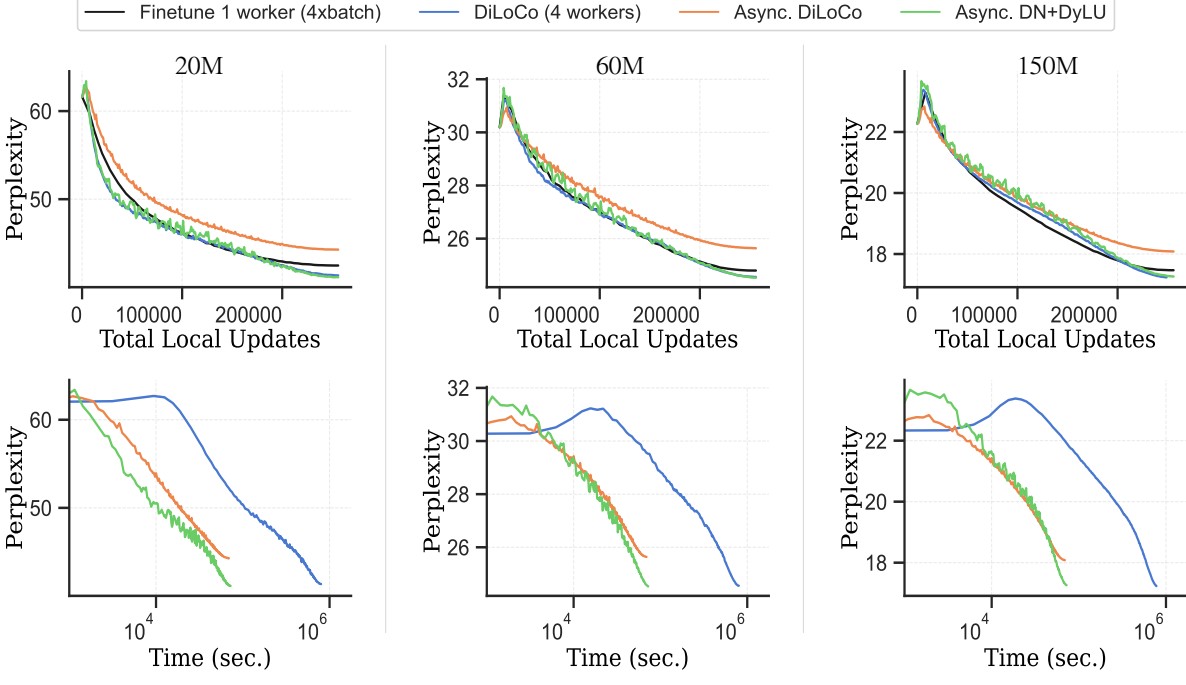

*Figure 11.* Varying the model size.

## C. A Minimal Toy Example

For the convenience of future research and quick prototyping of new ideas, we present a minimal toy example that replicates the observed optimization challenge in asynchronous Local-SGD (See Figure **??**).[5]

# D. Implementation Details

| Hyperparameter | Value |
|---|---|
| Inner learning rate | 0.1 |
| Final inner learning rate | 0.0, **0.000001**, 0.0002 |
| Number of warmup steps | 0, **1,000** |
| Weight decay | 0.1 |
| Batch Size | 128, 512 |
| Sequence length | 256 |
| Outer Optimizer | SGD, SGDM, Nesterov, Adam, **delayed momentum SGD** |
| Inner Optimizer | **SGD**, AdamW |
| Outer learning rate | 0.03, 0.3, **0.1, 0.7** |
| Async soup weight | 0.125, 0.25, 0.5, **1.0** |
| Async soup method | **constant**, polynomial, svrg |
| Delay period | **4**, 8, 16 |
| Communication frequency $H$ | **50**, 100, 150 |
| Number of pretraining steps | 24,000 |

*Table 5.* **Optimization Hyperparameters** evaluated during in this work. Chosen values for the main experiments are highlighted in bold.

**Network Architecture**  We displayed in Table 6 the architectural difference between the 20M, 60M, and 150M models. They are all transformer decoder-only, based on the Chinchilla family (Hoffmann et al., 2022).

**Training Dataset**  We consider a language modeling task on the C4 dataset, a dataset derived from Common Crawl (Raffel et al., 2020). The total number of steps is set to 88,000 for all models, with 24,000 steps of pre-training done without any federated learning methods, akin to *post Local-SGD* (Lin et al., 2020).

**Hyperparameters**  In Table 5, we outline the optimization hyperparameters considered for this study.

**Inner Optimizer States**  Following Douillard et al. (2023), in all experiments, when worker B picks up the data shard worker A just finishes training on, we reset the `AdamW`'s optimizer state. In other words, each local worker-side training is an independent training process with a new optimizer, and only the learning rate is adjusted according as described in Section 3.

# E. Additional Pseudocode for the Asynchronous Training Pipeline

In this section, we provide the pseudocode for the train() and get_worker() functions in Algorithm 1.

---

**Algorithm 4** train() in Algorithm 1.
1: **Require:** Available workers $\mathcal{W}$
2: **Require:** Current server model $\theta$
3: **for** $w \in \mathcal{W}$ **do**
4:     Sample shard $\mathcal{D}'$ for $w$ (Eq. 2).
5:     $w$.local_updates = DyLU($\mathcal{D}'$) (Eq. 6).
6:     Decide lr schedule ($w$.lr) (Eq. 3).
7:     $w$.update = train_worker($w, \mathcal{D}', \theta$).
8: **end for**

---

*Table 6.* **Model Configuration** for the three evaluated sizes. All are based on the transformer architecture, chinchilla-style (Hoffmann et al., 2022).

| Hyperparameter | 20M | 60M | 150M |
|---|---|---|---|
| Number of layers | 6 | 3 | 12 |
| Hidden dim | 256 | 896 | 896 |
| Number of heads | 4 | 16 | 16 |
| K/V size | 64 | 64 | 64 |
| Vocab size | | 32,000 | |

---

**Algorithm 5** get_worker() in Algorithm 1.

---

1:  **Require:** Workers $\mathcal{W}$
2:  **Require:** Grace period $\tau_{\text{grace}}$
3:  **Require:** Start of the grace period $\tau_{\text{sync}}$.
4:  **if** all workers in $\mathcal{W}$ are not done **then**
5:     **return** null
6:  **else**
7:     $w$ = earliest completed worker in $\mathcal{W}$.
8:     **if** $w$.completed_time $- \tau_{\text{sync}} \leq \tau_{\text{grace}}$ **then**
9:        **return** $w$
10:    **else**
11:       **return** null
12:    **end if**
13: **end if**

---

