# OpenReview forum: "Asynchronous Local-SGD Training for Language Modeling"
_ICML.cc/2024/Workshop/WANT — WANT@ICML 2024 Oral_

### Official Review · Reviewer_Z4Dd · 2024-06-10
**Promising Approach for Asynchronous Local-SGD for Language Model Pre-Training**

**Confidence:** 3

**Summary:**

This paper explores the use of asynchronous Local-SGD methods for training language models, integrating two novel techniques that outperform other tested asynchronous Local-SGD experiments: Delayed Nesterov momentum updates (DN) and Dynamic Local Updates (DyLU). They test their approach across various ablations for different techniques, model sizes ranging from 20M to 150M parameters, and a varying number of heterogeneous workers from 4 to 16.

**Strengths:**

- The paper includes comprehensive ablation studies with different asynchronous Local-SGD techniques.
- The introduction of Delayed Nesterov momentum updates (DN) and Dynamic Local Updates (DyLU) presents a novel approach to improving asynchronous Local-SGD performance.

**Weaknesses:**

- The methods were only demonstrated to scale up to models with 150 million parameters, raising questions about the scalability of the proposed techniques for larger models.

**Limitations:**

- A minimal toy example of the algorithm is provided, but there is no open-source implementation for running the main experiments of the paper, limiting the reproducibility of the results.

**Suggestions:**

- Adding details about which AI accelerators the models were trained on.
- It would be great in future work, if sufficient compute resources are available, to test this promising method on larger models to evaluate its scalability beyond 150 million parameters.

---

### Official Review · Reviewer_Z83L · 2024-06-12
**Important work for increasing the performance the of asynchronous fine-tuning of LLMs but missing reproducibility**

**Confidence:** 4

**Summary:**

The present work presents an asynchronous Local-SGD variant, that is helpful in distributed training settings of LLMs where communication between workers and heterogeneity in terms of computation speed of workers become the bottleneck. Empirical evaluations show that plain asynchronous Local-SGD variants converge slower than their synchronous counterparts. To address this problem, the paper introduces two momentum adaption techniques. One is based on strategically scheduling the updates to the momentum parameters while the other proposes an adjustment of the number of local training steps in relation to the speed of the worker. The results show the proposed method in most cases achieves better performance across different model sizes and number of workers, compared to related baselines.

**Strengths:**

- The idea to strategically delay the Nestov momentum updates in combination with an adjustment of the speed of the worker is novel and interesting, as it combines aspects of the computational efficiency (speed of the device) with aspects of the learning process (optimizer updates)
- The empirical evaluations show the method to outperform the DiLoCo baseline in most cases
- Different ablation studies in regard to model size, number of workers, and (hyper-) parameters of the momentum scheduling bolster the findings
- The idea to assign a learning rate scheduler to each local data shard is good.
- Results improve with the level of heterogeneity of the workers (Table 1), which is promising

**Weaknesses:**

- The focus of this work is on using the proposed method for the fine-tuning of LLMs. However, the largest potential for asynchronous SGD methods is to be realized during pretraining
- Evaluations are only performed up to a maximum number of 16 workers, which might be too small to see large benefits of asynchronous training settings

**Limitations:**

- As the authors also state in their limitations section, the benefits of using the Local-SGD approach begin to decline at larger scale, which can become an issue when moving to larger datasets in the future.
- The authors only release toy-problem code of their work, so the reproducibility of the results is very limited

**Suggestions:**

- If possible, evaluate the performance on more workers
- Release of the source code to foster adaption in the community and replication of the results

---

### Official Review · Reviewer_gr3e · 2024-06-14
**Extensive study of async gradient optimization; solid experimentation and ablation study.**

**Confidence:** 4

**Summary:**

The paper extensively study various pairs of local and global optimizer in async gradient training setup. The main empirical observation is that a momentum itself, its updates, and relations between momentum in inner and outer optimizers have a crucial role on convergence speed. The authors proposed two solutions to "synchronize" momenta and carried out solid experiments and ablations study to support authors' findings.

**Strengths:**

+ Interesting empirical observation which facilitates and motivates further research in this area.
+ Solid experiments and ablation study.

**Weaknesses:**

+ Missing theoretical analysis. However, I thinks that it is minor drawback and is promising direction for further research.
+ Template placeholder in the page header.
+ Bibliography should be reviewed and actualized:  capitalization of titles,
  missing publication dates, journals conferences, etc (the Petals paper
  published at ACL https://aclanthology.org/2023.acl-demo.54/).
+ Missing table of content in hypertext markup.

---

### Decision · Program_Chairs · 2024-06-18

**Decision:**

Accept (Oral)

**Comment:**

We thank the authors for their time and contribution to WANT and we are pleased to share that after the reviewing process the paper has been accepted. Congratulations! We encourage the authors to consider reviewers' feedback for the improvement of the camera-ready version. We hope to see you in person at the workshop and brainstorm on efficient training research together!